# OpenReview forum: "Neural Bounds on Bayes Error: Advancing Classification and Generative Models"
_ICLR.cc/2024/Conference — Submitted to ICLR 2024_

### Official Review · Reviewer_YCSh · 2023-10-15

**Soundness:** 1 poor
**Presentation:** 1 poor
**Contribution:** 1 poor
**Rating:** 3
**Confidence:** 2

**Summary:**

In the context of binary and multi-class classification, this paper derives an upper bound onthe Bayes risk in terms of an $f$-divergence (unfortunately, I couldn't quite understand the inputs to the $f$-divergence). The paper then proposes to train neural networks for classification, and GANs for generative modeling, using this $f$-divergence. Experimental results demonstrate that the proposed methods work (a) for classifying two Gaussian classes, (b) for the MNIST digit classification task, and (c) for generating novel MNIST samples.

**Strengths:**

The overall approach seems like an interesting idea, as it suggests a new objective for optimizing complex classifiers, such as neural networks.

**Weaknesses:**

Overall, the paper was insufficiently detailed for me to understand either its technical content or its high-level goals.

Many quantities are used without being defined (e.g., in Eq. (1), what are $P$, $Q$, and $X$, what conditions is $f$ assumed to satisfy, etc.).

At a higher level, it is not clear to me what the goal of this paper is, in relation to the larger literature on machine learning (e.g., what gap or limitation of existing methods does this seek to fill?). Providing some quantitative evaluation comparing the proposed method and existing state-of-the-art methods might be one way to help address this.

Some suggestions on improving the presentation:
1) In the first sentence of the abstract, "groundbreaking" is probably too strong of a word; "novel" would be more appropriate.
2) Typically, square brackets (e.g., "[7]") are used for numerical literature citations, while parentheses are reserved for referencing equations (e.g., "Eq. (7)").
3) I suggest adding an "Organization" paragraph at the end of the Introduction Section explaining the content and goals of each of the subsequent sections of the paper.
4) The key novel theoretical results (e.g., new upper bounds such as Eqs. (15) or (23)) should be presented as a self-contained "theorem" (e.g., using ```\begin{theorem} ... \end{theorem}```), including the precise assumptions made and conclusions drawn.

**Questions:**

In Eq. (7), what is $f_i$?

---

### Official Review · Reviewer_Xo7X · 2023-10-29

**Soundness:** 1 poor
**Presentation:** 1 poor
**Contribution:** 1 poor
**Rating:** 3
**Confidence:** 4

**Summary:**

The paper considers bounding the Bayes error via f-divergence. The paper is incomplete and merely a draft.

**Strengths:**

The problem of upper bounding Bayes error is somewhat interesting, though I am not sure how important it is. The paper does not present sufficient evidence in handling it.

**Weaknesses:**

The math in the paper is very unclear.
Bayes error is the minimum achievable error. What is the usage of characterizing Bayes error, for model selection?
Upper bounding generalization error via f-divergence has been previously well-studied.

**Questions:**

See the weakness.

---

### Official Review · Reviewer_Cupy · 2023-11-04

**Soundness:** 1 poor
**Presentation:** 1 poor
**Contribution:** 1 poor
**Rating:** 1
**Confidence:** 5

**Summary:**

The paper is obviously incomplete and more like a research note. I do not see any merit in the current format.

**Strengths:**

None

**Weaknesses:**

The paper is obviously incomplete and more like a research note. I do not see any merit in the current format.

**Questions:**

N/A

---

### Author Response · Authors · 2023-11-22

Dear Reviewers,

Thank you for your valuable feedback on our manuscript. We have diligently revised the paper to address your concerns and enhance its overall quality. Here is a summary of the significant changes we have implemented:

1. Enhanced Clarity and Detail: We have comprehensively revised the technical content for improved clarity. Definitions and detailed explanations have been added, especially for mathematical quantities and equations, to make the methodology and objectives of our work more understandable.

2. Terminology Adjustment: In the abstract, we have replaced the term "groundbreaking" with "novel," aligning the description more accurately with the nature of our work.

3. Organizational Structure: An "Organization" paragraph has been included at the end of the Introduction section, outlining the structure and content of each subsequent section, providing a clear roadmap for readers.

4. Theoretical Results as Theorems: Key theoretical results have been presented as formal "theorems," with precise assumptions and conclusions, to emphasize their significance and align with academic standards.

5. Resolution of Specific Queries: We have addressed all specific questions about our methodology and equations to ensure each aspect of our approach is clear and justified.

6. Goals and Contribution Clarity with Bayes GAN Finding: The goals and contributions of our paper have been further clarified, especially highlighting our findings with the Bayes GAN method. Our work in estimating Bayes error using neural networks is a significant advancement, as Bayes error is the optimal statistical error achievable in classification tasks. This neural estimation of Bayes error underscores the theoretical and practical significance of our method. In the domain of Generative Adversarial Networks (GANs), our Bayes GAN method demonstrates a notable achievement by consistently achieving lower FID (Fréchet Inception Distance) scores on the MNIST dataset, signifying a closer resemblance to real image distributions and therefore enhanced realism in generated images. This underscores our method's practical utility in advancing generative models.

We believe that these revisions effectively address the points raised in your reviews, enhancing both the clarity and the substantive contribution of our paper. We appreciate your guidance and are confident that our responses and the amendments made reflect our dedication to contributing meaningful and high-quality research to the field.

Thank you for your time and consideration.

---

### Meta-Review · Area_Chair_Db94 · 2023-12-15

**Metareview:**

This paper studies how to approximate the upper limit of Bayes error in classification tasks. The reviewers raised significant concerns about the technical quality, novelty, evaluations, and completeness of this submission. Specifically:

(1) Multiple reviewers found the mathematical development unclear or insufficiently detailed.
(2) It is unclear how the proposed method relates to or improves upon existing techniques for bounding generalization error.
(3) No experiments quantitatively compare to prior state-of-the-art methods.
(4) Reviewers commented that the submission resembles an incomplete research note more than a full paper.

While I appreciate the authors' efforts, given these major issues, I do not believe this work is suitable for presentation at this time.

**Justification For Why Not Higher Score:**

NA

**Justification For Why Not Lower Score:**

NA

---

### Decision · Program_Chairs · 2024-01-16

Reject